# Lactoferrin Stimulates Chondrogenesis and Promotes Healing of the Auricular Elastic Cartilage

**DOI:** 10.3390/ijms26051956

**Published:** 2025-02-24

**Authors:** Anastasiia D. Kurenkova, Natalia B. Serejnikova, Sofia A. Sheleg, Alexey L. Fayzullin, Nikolai E. Denisov, Alexandra V. Igrunkova, Elena R. Sadchikova, Artem A. Antoshin, Peter S. Timashev

**Affiliations:** 1Institute for Regenerative Medicine, Sechenov First Moscow State Medical University (Sechenov University), 8-2 Trubetskaya St., 119991 Moscow, Russia; serezhnikova_n_b@staff.sechenov.ru (N.B.S.); fayzullin_a_l@staff.sechenov.ru (A.L.F.); antoshin_a_a@staff.sechenov.ru (A.A.A.); timashev_p_s@staff.sechenov.ru (P.S.T.); 2Department of Human Anatomy and Histology, N.V. Sklifosovsky Institute of Clinical Medicine, Sechenov First Moscow State Medical University (Sechenov University), 11, Building 10, Mokhovaya St., 125009 Moscow, Russia; 3Institute of Gene Biology, Russian Academy of Sciences, 34/5 Vavilov St., 119344 Moscow, Russia; e.r.sadchikova@gmail.com; 4World-Class Research Center “Digital Biodesign and Personalized Healthcare”, Sechenov First Moscow State Medical University (Sechenov University), 8-2 Trubetskaya St., 119048 Moscow, Russia

**Keywords:** lactoferrin, auricular chondrocytes, elastic cartilage, cartilage regeneration, ear reconstruction

## Abstract

Ear reconstruction surgeries for congenital deformities and trauma are common, highlighting the need for improved cartilage regeneration. Lactoferrin (LF), a natural and cost-effective protein, is promising due to its anti-inflammatory, antimicrobial, and prochondrogenic properties. This study investigates the effects of LF on the viability, proliferation, and chondrogenesis of rabbit auricular chondrocytes. For in vitro studies, auricular chondrocytes were cultured for three passages, after which 3D pellets were formed. LF significantly increased chondrocyte metabolic activity by 1.5 times at doses of 10 and 500 μg/mL. At passage 3, LF at concentrations of 10 and 100 μg/mL increased cell proliferation rates by 2- and 1.5-fold, respectively. Immunohistochemical staining of the pellets demonstrated that LF at 10 μg/mL increased the amount of sex-determining region Y-Box Transcription Factor 9 (Sox9)+ cells by 30%, while at 100 μg/mL, it doubled the type II collagen deposits. For in vivo studies, a rabbit ear defect model was utilized. On post-operative day 60, the LF-treated group exhibited more mature cartilage regeneration, with a higher density of elastic fibers. By day 90 post-surgery, LF application led to the restoration of normal elastic cartilage throughout the defect. These findings suggest that LF promotes auricular chondrocytes chondrogenesis and could be beneficial for tissue engineering of the elastic cartilage.

## 1. Introduction

The auricle is vulnerable to injuries due to its location and the very thin layer of skin covering it. Deformation of the pinna can arise from hereditary conditions such as microtia and anotia or can be acquired due to animal bites, head trauma, burns, and other causes. These deformities may lead to cosmetic defects and impact sound localization [1]. Therefore, the issue of auricular reconstruction remains relevant.

The elastic cartilage, which constitutes the auricle, is an avascular tissue with an extracellular protein matrix reinforced by a three-dimensional network of collagen fibers. It has limited regenerative capacity due to the sparse distribution of highly differentiated, non-dividing chondrocytes, slow matrix turnover, and a low reserve of progenitor cells [2]. For decades, auricular reconstruction has involved forming an ear scaffold from autologous costal cartilage [1]. This technique yields satisfactory results due to the similar properties of the cartilages, which allow for the creation of a durable, biocompatible scaffold. However, it has drawbacks, such as cartilage transplant deformation over time, the risk of pneumothorax, and infection due to cartilage harvesting [3].

Nowadays, tissue engineering and the creation of artificial cartilage from the patient’s cells have become more preferable, modern, and safer methods. These approaches avoid unnecessary surgical interventions and the use of cartilage analogs [4]. Various tissue engineering approaches are employed, including different scaffolds [5], 3D bioprinting [6,7,8], cell-loaded hydrogels [9,10], and scaffold-free techniques [11,12,13]. Nevertheless, challenges include low tissue cellularity [14], the need for vascularization of the skin and cartilage, limited cell proliferation, risks of oncogenicity and abnormal differentiation, thermal and mechanical forces, and mechanical and immunological complexities in integration with the host tissue [15]. Therefore, modifications to existing techniques or the development of new methods are necessary to overcome these challenges.

Since the mid-20th century and to this day, researchers have been interested in the pro-regenerative properties of the iron-binding protein lactoferrin (LF) [16]. Studies have shown that LF exerts anti-inflammatory and anti-apoptotic effects and stimulates cell proliferation, positively influencing the regeneration of articular hyaline cartilage. In human articular chondrocytes, LF has been shown to reduce apoptosis [17] and stimulate chondrocyte proliferation [18], as well as decrease the expression of catabolic matrix metalloproteinases (*MMP-1*, *MMP-3*, and *MMP-13*), destructive cytokines (interleukins(*IL*)*-1β* and *IL-6*), and inflammatory mediators (inducible nitric oxide synthase *iNOS*, cyclooxygenase-2) during IL-1β-induced inflammation [19,20]. LF can induce the protective expression of anti-inflammatory cytokines *IL-4*, *IL-10* [19], and *IL-11* [21], and activate the expression of bone morphogenetic protein *BMP-7*, exerting a pro-chondrogenic effect [22].

In vivo models have shown that LF improves bone condition and reduces the expression of catabolic metalloproteinases in a mono-iodoacetate-induced temporomandibular joint osteoarthritis model in rats [23]. In therapy, LF can be used in combination with other anti-inflammatory drugs, such as dexamethasone, as it can reduce the pro-apoptotic effect on articular chondrocytes [24].

The effects of LF have also been studied in the treatment of intervertebral disc diseases. LF prevents the decrease in proteoglycan synthesis by nucleus pulposus chondrocytes and reduces the expression of catabolic metalloproteinases and pro-inflammatory factors *iNOS* and *IL-6* during IL-1-induced inflammation [25]. Additionally, its synergistic effect with pro-chondrogenic BMP-7 has been noted in nucleus pulposus chondrocytes [26].

Given the pronounced anti-inflammatory and pro-regenerative effects of LF on chondrocytes, we hypothesize that it can stimulate the regeneration of elastic cartilage, making it applicable in reconstructive surgery and tissue engineering of the auricle.

## 2. Results

### 2.1. LF Increases Viability and Proliferation of Auricular Chondrocytes

The effect of LF on the viability of auricular chondrocytes was evaluated for cells at passage 1 using the Alamar Blue assay. Various concentrations of LF were utilized to establish a dose-dependent effect. It was shown that short-term cultivation with LF at a concentration of 0.1 µg/mL significantly increased chondrocyte viability, although such a noticeable effect was not observed at other doses (Figure 1B). During prolonged cultivation, the effects of LF were more pronounced: at concentrations of 10 and 500 µg/mL, it increased the metabolic activity by 1.5 times (Figure 1C).

To evaluate LF’s long-term impact, we assessed its effects on proliferation. Cells were expanded over three passages, and the fold increase in cell number was recorded as the proliferation rate. The general appearance of auricular chondrocytes in culture is presented in Figure 1D. The cells were cultured with LF added to the medium at four different concentrations. In the first two passages, LF did not significantly affect cell proliferation, which was generally low. However, in the third passage, a sharp increase in the proliferation rate was observed, with LF at concentrations of 10 and 100 µg/mL significantly stimulating this process by 2 and 1.5 times, respectively (Figure 1E).

Thus, LF can stimulate the metabolic activity of cells and their proliferation, although an appropriate dosage needs to be selected. A broad range of LF concentrations, from 10 to 200 µg/mL, has been used in studies of articular cartilage [16,17,18,19,20,21,22,23,24,25,26]. For our experiments, 10 µg/mL was identified as the minimum required concentration, as it provided both maximal proliferation enhancement and cell viability. Therefore, in further experiments on the effect of LF on chondrogenic differentiation, cells grown in a medium supplemented with 10 µg/mL of LF were used.

### 2.2. LF Stimulates Chondrogenesis in 3D Pellets Culture of Auricular Chondrocytes

To evaluate the ability of LF to stimulate chondrogenesis in vitro, we used a 3D pellets culture model. Pellets were formed from cells that had been expanded in a medium containing 10 µg/mL of LF and then cultured for 3 weeks in a medium containing three different concentrations of LF: 10, 100, and 1000 µg/mL. Histological analysis with orcein staining showed the presence of elastic fibers in the extracellular matrix in all groups (Figure 1H–K). Mallory staining did not reveal a significant amount of proteoglycans (Figure 1L–O), which is consistent with the composition of elastic cartilage in the ear. It is noteworthy that over three weeks in pellets culture, chondrocytes did not form a dense extracellular matrix in any pellets, which was easily identifiable by histological staining. Nevertheless, immunohistochemical staining demonstrated the presence of type II collagen in all groups, although its localization was clustered (Figure 1P–S). Collagen quantification based on fluorescence levels revealed that in the group treated with 100 µg/mL LF, values were twice as high as in the control group. Increasing LF concentration to 1000 µg/mL resulted in a decrease in type II collagen levels, making them indistinguishable from the control (Figure 1F). Staining for the chondrocyte marker Sox9 (sex-determining region Y-Box Transcription Factor 9) showed a high percentage of positive cells in all samples (Figure 1T–W). Quantitative analysis of the percentage of Sox9 positive cells showed a significant 30% increase in the group with 10 µg/mL LF (Figure 1G). Increasing the concentration of LF led to a decrease in the percentage of Sox9-positive cells, although this effect was not statistically significant.

Thus, it can be concluded that LF can stimulate chondrogenesis, although dose optimization is required. In our experiments, the optimal LF concentrations were 10 and 100 µg/mL, as they resulted in the highest levels of chondrogenic markers Sox9 and type II collagen.

### 2.3. LF Promotes Healing of Cartilage Defects in Rabbit Ear

To evaluate the ability of LF to stimulate elastic cartilage regeneration in vivo, we utilized a rabbit ear defect model (Figure 2A). Macroscopic analysis showed that at post-operative day 30 (POD30), most defects in the control and experimental groups were epithelialized, characterized by a central retraction that pulled and deformed the surrounding tissues, with little difference in color and consistency between them. At POD60, most defects in both groups were completely epithelialized, with newly formed tissue exhibiting a smooth, flat surface, gray color, and firm-elastic consistency (Figure 2B,C). By POD90, complete normalization of wound color was observed, and the newly formed tissue had the same density as the surrounding tissues. Macroscopic scoring showed no statistically significant differences between the groups (Figure 2D).

At POD30, histological analysis revealed immature fibrous cartilage tissue formed near the edges of the intact cartilage plate in both groups. In the control (Figure 2E), the regenerating chondrocytes were predominantly small and round, surrounded by collagen fibers, with only some cells having lacunae. Dense immature fibrous tissue composed of fibroblasts, blood vessels, and thick bundles of collagen fibers filled the space between the edges of the regenerating cartilage plate. In the LF group (Figure 2F), larger cartilage regenerates with a high density of proliferating chondrocytes were observed compared to the control. The volume of fibrous tissue between the edges of the cartilage regenerates was reduced. Orcein staining revealed only small areas of regenerating cartilage with few elastic fibers in the control group (Figure 2G), whereas extensive layers of elastic cartilage formed in the LF group (Figure 2H).

By POD60, cartilage plate regeneration significantly accelerated, leading to a marked increase in the area of fibrous cartilage. Regenerating chondrocytes increased in size and became more dispersed. In the control, less mature regions of cartilage regenerate with small, numerous chondrocytes (Figure 2I), and low content with uneven distribution of elastic fibers continued to be observed (Figure 2K). In the LF group, compared to the control, more mature cartilage regenerate areas with large round chondrocytes and large lacunae (Figure 2J) predominated, uniformly surrounded by numerous elastic fibers (Figure 2L).

By POD90, the structure of the cartilage regenerated in both groups resembled normal elastic cartilage (Figure 2M–P), but only in the LF group did the cartilage plate regenerate along the entire length of the defect in most samples. The LF group achieved complete regeneration of the cartilage plate with restoration of the elastic cartilage type, as confirmed by orcein staining (Figure 2O).

Morphometric analysis of cartilage regeneration quality (Figure 2Q,R) showed that in the LF group, the sizes of cartilage regenerated with a high degree of maturity and the content of elastic fibers were significantly higher than in the control at POD60. No statistically significant differences were found between the groups at other time points, although a trend was observed.

Thus, the obtained data suggest that LF positively influenced the organotypic regeneration of ear cartilage compared to the control group. The extent of highly mature cartilage regenerates with elastic fiber content increased linearly over time and was maximal following LF treatment at POD60.

## 3. Discussion

A significant amount of research has been dedicated to investigating the effects of LF on the hyaline cartilage of joints. It has been demonstrated that LF possesses anti-inflammatory [19,20,21], anti-catabolic [17,19], and prochondrogenic effects [20,22]. However, such studies have not been previously conducted on the elastic cartilage of the ear.

In this study, we demonstrated the ability of LF to stimulate the metabolic activity of auricular chondrocytes. Previous studies have shown similar effects of LF on articular chondrocytes at comparable doses [18]. It is important to note that many metabolic assays, including Alamar Blue, are not suitable for evaluating the long-term effects of substances due to their intrinsic impact on metabolism [27,28]. Therefore, for periods exceeding 24 h, we assessed the effect of LF by evaluating cell proliferation. Our research demonstrated that LF does not affect the division of auricular chondrocytes in early passages; however, its effects become more pronounced by the third passage, i.e., by the third week of cultivation.

Interestingly, the stimulatory effect of LF was concentration-dependent. When examining its effects on metabolic activity and cell proliferation, the most effective concentration range was 10–500 µg/mL. For subsequent experiments, cells were expanded at the minimally effective concentration of 10 µg/mL. In the case of chondrogenic differentiation, the most pronounced increase in Sox9-positive chondrocytes was observed at 10 µg/mL. Indirect quantification of collagen deposition via fluorescence revealed that the most significant extracellular matrix deposition occurred at 100 µg/mL. At 10 µg/mL, collagen levels were higher than the baseline but did not reach statistical significance (*p* = 0.07). It is worth noting that we did not perform PCR analysis due to limitations in the amount of biological material from animals; however, earlier studies using chondrocytes show similar patterns between the expression of extracellular matrix proteins and their content in tissue [29,30,31].

Notably, increasing the LF concentration to 1 mg/mL did not produce significant effects in either 2D or 3D cultures. The literature indicates that low doses of LF can stimulate metabolism in intestinal epithelial cells, whereas higher doses exhibit pro-apoptotic effects [32]. In our study, 1 mg/mL of LF did not exert such effects; however, the protein activity decreased with increasing concentration. Thus, we identify the 10–100 µg/mL range as the optimal concentration interval for stimulating proliferation, chondrogenesis, and metabolic activity in auricular chondrocytes, with 10 µg/mL being the minimally effective concentration.

Earlier studies have shown that LF’s mechanism for stimulating metabolic activity has some homology with transforming growth factor (TGF)-β1 [18]. Furthermore, LF appears to partially mediate chondrogenesis via the Smad2/3-Sox9 signaling pathway [33]. Additionally, LF can also stimulate the BMP signaling pathway to enhance chondrogenesis, as shown for various types of chondrocytes [22,26]. These effects involve Smad signaling, whose activity is multifactorial and difficult to predict. Apart from ligand concentration, factors such as the balance of specific mediators, negative feedback mechanisms, receptor endocytosis, and other elements may influence the final outcome [34]. It should be noted that this signaling pathway is not the only molecular target of LF; it can interact with damage-associated molecular patterns (DAMP) and pathogen-associated molecular pattern (PAMP) receptors [35] and modulate the nuclear factor kappa B (NF-kB) pathway [36], as well as activate extracellular signal-regulated kinase (ERK), mitogen-activated protein kinase (MAPK), and Akt signaling pathways [19]. Thus, from a molecular perspective, its action on cells is quite complex. We hypothesize that this complexity may contribute to a nonlinear dose–effect relationship, making it difficult to directly extrapolate findings from one cell culture type to another. Nevertheless, our identified optimal range (10–100 µg/mL) aligns with the concentrations that influence articular chondrocytes, typically 10–200 µg/mL [16,17,18,19,20,21,22,23,24,25,26]. However, the optimal dose of LF should be empirically determined to avoid adverse effects.

Thus, LF may be a highly promising supplement to the culture medium, promoting chondrogenesis in elastic cartilage equivalents. This is particularly important for the rapidly advancing field of bioprinting and the cultivation of tissue-engineered constructs from various cells [7].

In in vivo experiments on rabbit ear cartilage healing, LF also demonstrated its pro-regenerative properties. It is known that the rabbit ear cartilage plate can fully regenerate under certain conditions [37]. The size, type (volume), and location of the defect play significant roles in the healing of full-thickness defects in auricular cartilage in rabbits [38,39]. The outcome of the wound healing can be either the complete restoration of the tissues of the ear or the partial replacement of them with a scar. The factors controlling the regeneration of rabbit ear cartilage are understudied [40].

The results of our study demonstrated that LF had a beneficial effect on organotypic regeneration of ear cartilage, accelerating the growth and maturation of regenerating cartilage compared with the control group. It is important to note that elastic cartilage formed, replacing fibrous cartilage tissue over the course of regeneration. The number of elastic fibers increased linearly depending on time. In the LF’s group, compared to the control, elastic fibers formed faster (~1.5 times) in newly formed cartilage at POD60. It is well known that LF stimulates the release of cytokines and growth factors [41] that could modulate elastogenesis [42]. Also, this effect may be mediated by LF’s well-known anti-inflammatory [36] and antimicrobial properties [43].

So, LF may be responsible for the additive stimulatory effect on the formation of elastic fibers. As a result, in LF’s group in the regenerating cartilage, large lacunae were surrounded by numerous elastic fibers, and dystrophic changes were completely absent. The morphological characteristics of the tissue were as close as possible to those of the normal elastic cartilage tissue.

Our study has several limitations that should be addressed in future research. First, although we demonstrated the overall pro-chondrogenic effect of LF in both in vitro and in vivo experiments, we did not explore its molecular aspects in depth. As previously described, the mechanism of LF action may involve modulation of TGF-β signaling pathway activity. Gene expression analysis or quantification of specific molecules in this cascade could provide further insights into both the stimulation of proliferation and the effects of LF on chondrogenesis markers.

In the in vivo experiments, we focused exclusively on the microscopic analysis of the morphological aspects of healing of the auricular elastic cartilage. Epimorphic regeneration is a complex process, and its understanding is crucial for the future of regenerative medicine. The TGF-β signaling pathway and the B-cell lymphoma 2 (Bcl-2) family of proteins play a significant role in epimorphic regeneration [44]. LF is known to act on TGF-β1 receptors [18], and in osteoblasts, it has been shown to inhibit Bcl-2 while stimulating autophagy [45]. Thus, the pro-regenerative effects of LF on the auricle may be at least partially mediated by these signaling pathways.

Secondly, in our study, we used only the scoring for a qualitative assessment of morphological features of cartilage regeneration (chondrocyte density, elastic fiber content). Semi-automatic image analysis would enable us to make precise measurements and evaluations of cellular and tissue components of regenerative tissue [46].

Thirdly, this study was conducted on rabbits, which may influence the results. For future translational research, it is crucial not only to replicate our findings but also to consider additional factors. For instance, we used only the resazurin-based assay for the primary evaluation of cytotoxicity and viability, as its results correlate well with those of other metabolic tests [47]. However, for human applications, a broader panel of assays is required. Additionally, while we identified an optimal range of LF concentrations for future studies, selecting a dose for human application should consider a more extensive set of markers related to elastic tissue and regeneration, including their expression levels.

Despite the similarity of structure and functional processes, the rabbit auricular elastic cartilage is not fully comparable with the human cartilage due to some of its unique features. The rabbit ear has a relatively large “supply” of easily available cartilage and it is a well-accepted model for histological, biochemical, and biomechanical analyses of cartilage repair [40,48]. The results of studies of rabbit ear elastic cartilage usually correlate with the structure of human cartilage tissue, its biochemical composition, and mechanical properties [49], and the metabolism in immature and mature rabbit and human auricular chondrocytes is comparable [50]. Despite significant similarities, cartilage thickness, cell density, the presence of adipochondrocytes as a unique cell type, the distribution of different collagen types in the matrix, the total glycosaminoglycan content, and chondrocyte gene markers of rabbit auricular cartilage do not fully correspond to those in human auricular cartilage [51]. Rabbit and human chondrocytes can differ in cell behavior, growth, and extracellular matrix production, which can affect the healing of the auricular elastic cartilage [52]. Differentiation and proliferation are also affected by the surrounding microenvironment, including soluble factors and mechanical loading [53,54]. Caution should be exercised when interpreting data from this animal model in translational research that aims at human application.

Thus, within the framework of this study, we have confirmed the hypothesis that LF can influence the regeneration of elastic cartilage. Future translational studies employing a broader range of methods, such as comprehensive cytotoxicity assessment, in situ hybridization, and spatial transcriptomics, will provide insights into the fundamental mechanisms of organotypic regeneration and the activation of these mechanisms in differentiated tissues of adult organisms.

## 4. Materials and Methods

### 4.1. Animals

All the animal experiments were approved by the bioethics committee of the Sechenov University (No. 13–22 from 22 June 2022, Moscow, Russia). Experiments were conducted in accordance with Directive 2010/63/EU of the European Parliament and the Council, and with FELASA guidelines and recommendations. The study utilized Chinchilla rabbits (males, 2–2.5 kg). The rabbits were housed under standard vivarium conditions, each in an individual pet cage, and were provided with a complex pelleted laboratory diet along with constant access to water.

### 4.2. Auricular Chondrocytes Isolation

For cell isolation, rabbit ears were harvested post-euthanasia using Zoletil 100 (Vibrac, Carros, France; 60 mg/kg) followed by cervical dislocation. The skin was separated using a blade, and the obtained cartilage was washed in PBS solution and then in DMEM (Gibco, Waltham, MA, USA, 41966029) containing gentamicin 20 µg/mL. The cartilage was then cut into smaller fragments, homogenized, and incubated for 15 h at 37 °C in a 0.1% collagenase type II (Worthington Biochemical Corp., Lakewood, NJ, USA, LS004177) solution in DMEM supplemented with 20 µg/mL gentamicin. After incubation, the cells were passed through a 100 µm cell strainer and seeded into a culture dish.

### 4.3. Cell Culture

The overall in vitro experimental scheme is shown in Figure 1A. Cell expansion was conducted over three passages in Dulbecco’s modified Eagle medium DMEM/F12 medium (Gibco, Waltham, MA, USA, 11320033) containing 10% fetal bovine serum (FBS) (Gibco, A3160801) and 10 µg/mL of gentamicin (Sigma, St. Louis, MO, USA, G1272-100ML). To study the effect of LF on proliferative activity, it was added to the medium at concentrations of 10, 100, and 1000 µg/mL, while control cells received an equivalent volume of phosphate-buffered saline (PBS). LF was provided by the Institute of Gene Biology, Russian Academy of Sciences, Moscow [55].

The proliferation rate was determined as the ratio of the number of cells at the end of the culture period to the number of cells initially seeded. Based on the experimental results, chondrogenic differentiation was performed only for the experimental group with the highest proliferation.

After the third passage, cells were collected and 3D pellet cultures were formed for chondrogenic differentiation, according to the protocol previously described [56]. Briefly, 500,000 cells were collected in tubes and centrifuged at 400× *g* for 7 min, and the pellet was maintained in culture conditions upon formation. Differentiation was conducted in chondrogenic medium (DMEM high glucose supplemented with 100 µM of 2-mercaptoethanol, 2 mM of sodium pyruvate, 0.35 mM of L-proline, 1% ITS + 3, 1% penicillin–streptomycin, 5 µg/mL of ascorbic acid, 0.1 µM of dexamethasone, and 10 ng/mL of TGF-β3) for 3 weeks. LF was also added to the medium at concentrations of 10, 100, and 1000 µg/mL, while control pellets received an equivalent volume of PBS.

### 4.4. Cell Viability Test

Cell viability was assessed at passage 1. Cells were seeded into 96-well plates and grown in an expansion medium until they reached 70–80% confluency. The medium was then replaced with a fresh medium containing Alamar Blue (Invitrogen, Waltham, MA, USA, DAL1025) and LF at concentrations of 0.01, 0.1, 1, 5, 10, 50, 100, 500, and 1000 µg/mL, or PBS in the control. For short-term viability assessment, cells were cultured for 4 h, as recommended by the manufacturer. We also attempted to assess longer-term effects and for this, cells were cultured for 17 h (overnight). Resorufin fluorescence was measured using a Victor Nivo spectrophotometer (PerkinElmer, Springfield, IL, USA). The fluorescence of the culture medium with added Alamar Blue was used as background values, which were subtracted from the fluorescence values of each well.

### 4.5. Surgery

For the surgery, the animals were anesthetized by intramuscular injection of Zoletil 100 (Vibrac, France; 6 mg/kg), supplemented with local anesthesia of the surgical site using a 0.5% solution of novocaine. Full-thickness defects were created using 6 mm diameter biopsy punches (Dermal Punch, Sterylab, Milan, Italy). Five punch holes were made in each rabbit’s ear, spaced 10 mm apart (Figure 1A). The wounds were dressed with Cosmopore bandages (Paul Hartmann, Heidenheim an der Brenz, Germany) and treated with a 3% hydrogen peroxide solution for three days.

On the 4th week post-surgery, the animals were divided into two groups (three rabbits each): a control group with no treatment and a treatment group where the peripheral wound areas were injected with 0.1 mL of a LF solution at a dose of 0.06 mg. The dosage was calculated based on the previous studies [16,57]. On days 30, 60, and 90 post-treatment (POD30, POD60, and POD90), the animals were euthanized by administering Zoletil 100 (Vibrac, Carros, France; 60 mg/kg). The defect areas, along with surrounding tissues approximately 2–3 mm from the edges of the original wounds, were excised and fixed.

### 4.6. Macroscopic Evaluation

Macroscopic evaluation of tissues surrounding the defect area was conducted at predetermined control points (POD30, POD60, and POD90). The visual assessment included the percentage of defect closure, defect density, defect color, and surface texture. A semiquantitative analysis using a scoring system was also performed (Table 1).

### 4.7. Histology

All samples were fixed in 10% neutral buffered formalin (BioVitrum, Saint-Petersburg, Russia, B06-001/L). After 3 weeks of cultivation, pellets were fixed for 3 h, followed by dehydration in 30% sucrose before embedding in OCT. Pellet sections of 16 µm thickness were obtained using a cryostat. Immunohistochemical staining for cartilage markers Sox9 (1:100, Sigma, HPA001758) and collagen II (1:200, Invitrogen, PA1-26206) was performed using a modified rabbit-on-rabbit staining protocol. For Sox9, antigen retrieval was performed by boiling it in citrate buffer, and for collagen II, boiled with 0.5% pepsin in 5 mM HCl for 20 min at 37 °C. Washing was performed with TBST (Tris-buffered saline with 0.1% Tween 20). Blocking was performed in a TBS buffer containing 0.1% Tween 20, 0.1% Triton X100, and 6% normal horse serum for 5 h. Incubation with primary antibodies was performed overnight at 4 °C, and with secondary antibodies (1:500, Jackson Immuno Research, West Grove, PA, USA, 711-165-152) for 1 h. Nuclei were stained with DAPI. Images were captured using an Olympus FLUOVIEW FV3000 confocal microscope (Tokyo, Japan). Staining was performed following the same protocol, and imaging was conducted under identical settings. Immunohistochemical image analysis was carried out using the ImageJ version 1.53e software. The percentage of Sox9-positive cells was determined by counting stained nuclei relative to all nuclei identified using DAPI, employing the ITCN (Image-based Tool for Counting Nuclei) plugin. Quantification of type II collagen fluorescent signals was performed using the Plot Profile function, with the methodology described in detail in [58].

Ear defect samples were fixed for 24 h and underwent standard processing. Briefly, samples were dehydrated using isopropyl alcohol (BioVitrum, 06-002/L) in an Epredia STP120 automated tissue processor (Thermo Fisher Scientific, Waltham, MA, USA). After dehydration, the samples were embedded into paraffin blocks (BioVitrum, Histomix, 247). Sections (4 μm thick) were prepared from the paraffin blocks using a microtome (Sakura Finetek Japan Co., Ltd., Tokyo, Japan), placed on slides, and dried in a thermostat at 37 °C for 24 h before staining.

Histological staining with hematoxylin and eosin (BioVitrum, 05-002), Mallory’s trichrome (BioVitrum, 21-032), and orcein (BioVitrum, 21-034) was performed according to the manufacturer’s protocol.

A Hamamatsu Nanozoomer S20 histological scanner (Hamamatsu, Japan) was used for slide imaging. An experienced pathologist described the following morphological findings: the distance between defect edges, epithelization, the characteristics of subepithelial tissue (regenerated dermis/granulation tissue/scar), the characteristics of cartilage plate regeneration (volume of granulation tissue, degree of cell differentiation, thickening of the perichondrium), and the presence of dystrophic changes in intact and regenerated cartilage.

### 4.8. Morphometric Analysis

Semi-quantitative evaluation of epimorphic regeneration and elastic fiber content was performed using a scoring system reported in a previous study [39] (Table 2 and Table 3). Morphometric analysis was conducted using NDP.view 2 version 2.5 software (Hamamatsu, Japan). For each rabbit, multiple sections were assessed to ensure the accuracy and reliability of the data. Serial sections were made for each rabbit with a step size of 50 µm, and for each defect, 5 sections were assessed and then averaged.

### 4.9. Statistical Analysis

Statistical analysis of experimental data was performed using GraphPad Prism 8 (GraphPad Software, Inc., La Jolla, CA, USA). The normality of the distribution was determined using the Shapiro–Wilk test. For chondrocyte viability and percent of Sox9+ cells analysis, the Brown–Forsythe ANOVA test was used, followed by multiple comparisons with a single control group using Dunnett’s test. For the assessment of proliferative activity, a mixed-effects model (REML) was employed, followed by Dunnett’s multiple comparisons test. Data are presented as mean and 95% confidence interval.

For morphometric analysis, the Mann–Whitney test was used for group comparisons within a single time point. Data are presented as median and interquartile range.

In all tests, the significance level was set at α = 0.05. Differences were considered statistically significant at *p* < α.

## 5. Conclusions

This study aimed to investigate the potential of LF as a bioactive factor for enhancing chondrogenesis in elastic cartilage and evaluating its effectiveness in an animal model of auricular cartilage repair. Our findings indicate that LF promotes chondrocyte viability, proliferation, and extracellular matrix deposition in a concentration-dependent manner, with optimal effects observed at 10–100 µg/mL. Notably, in vivo experiments revealed that LF-treated defects exhibited superior cartilage regeneration with more organized elastic fibers and restored structural integrity when compared to controls.

Beyond its relevance to auricular cartilage repair, these findings contribute to the broader understanding of LF’s possible therapeutic applications in regenerative medicine. Its low cost, ease of availability, and the small doses required for a comprehensive biological effect make it a promising candidate for clinical applications. LF can be used as a component of culture media for growing various tissue-engineered constructs and may also be applied locally to stimulate intrinsic regenerative mechanisms. However, future research should further explore LF’s potential for clinical translation, including its long-term effects and applicability in humans. In addition, several fundamental questions remain unanswered, particularly regarding the molecular mechanisms through which LF exerts its effects, necessitating further investigation.

Overall, this study provides a foundation for further exploration of LF as a promising agent in elastic cartilage regeneration, with implications for both clinical applications and fundamental cartilage biology research.

## Figures and Tables

**Figure 1 ijms-26-01956-f001:**
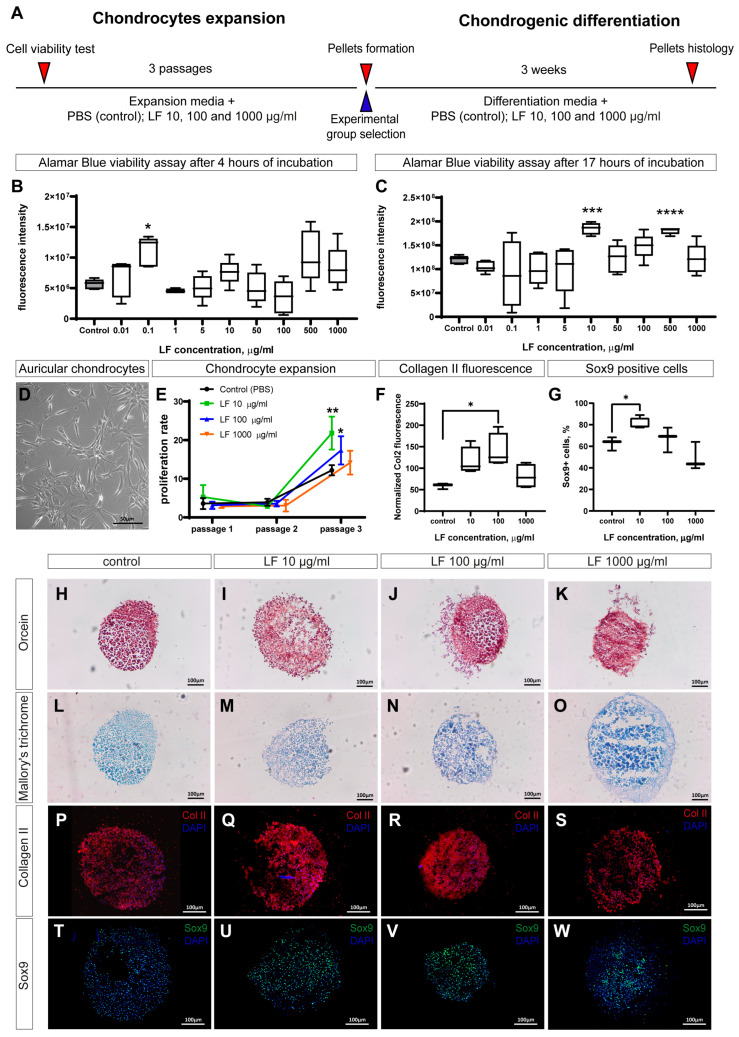
In vitro experiments. (**A**) Overall experimental scheme. (**B**) Viability assessment of auricular chondrocytes using various concentrations of LF after 4 h of cultivation (n = 5). (**C**) Viability assessment after 17 h of cultivation (n = 5). (**D**) General view of chondrocytes in culture (**E**) Proliferation rate of chondrocytes over three passages with different concentrations of LF (n = 6–10). (**F**) Collagen type II fluorescence normalized to pellet area (n = 4). (**G**) Percentage of Sox9-positive chondrocytes in formed pellets (n = 4). (**H**–**W**) Histological and immunostaining of pellets for key cartilage tissue markers: orcein staining for elastic fibers (**H**–**K**), Mallory staining for extracellular matrix (**L**–**O**), type II collagen staining (**P**–**S**), and Sox9 staining (**T**–**W**). LF—lactoferrin; PBS—phosphate saline buffer; Col II—collagen type II; Sox9—sex-determining region Y protein (SRY)-Box Transcription Factor 9. * *p* < 0.05, ** *p* < 0.01, *** *p* < 0.001, **** *p* < 0.0001.

**Figure 2 ijms-26-01956-f002:**
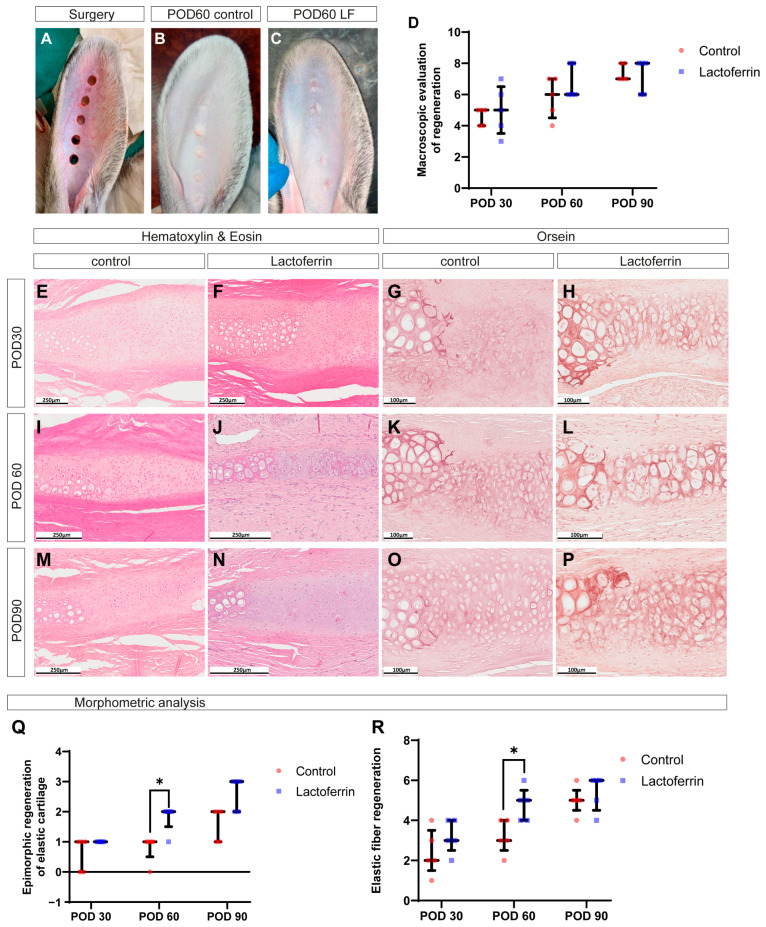
In vivo experiments. (**A**) Appearance of rabbit ear defect and its healing at post-operative day 60 in the control group (**B**) and the LF group (**C**). (**D**) Macroscopic assessment of regeneration. Staining of defect healing on post-operative day 30 (**E**–**H**), 60 (**I**–**L**), and 90 (**M**–**P**), with hematoxylin–eosin and orcein staining. Semi-quantitative morphometric analysis of obtained sections: epimorphic regeneration (**Q**) and elastic cartilage regeneration (**R**). LF—lactoferrin; POD—post-operative day. * *p* < 0.05.

**Table 1 ijms-26-01956-t001:** Scoring system for overall assessment of cartilage defect regeneration [39].

Score	Wound Consolidation	Density	Color	Surface Texture
0	≤25%	Entire defect dense	Red	Deformed with retractions and depressions
1	25–50%	Wound edges dense	Pale	Smooth and even
2	50–75%	-	Pigmented	-
3	˃75%	-	Normal	-

The maximum total score is 8.

**Table 2 ijms-26-01956-t002:** Scoring system for evaluating epimorphic regeneration of cartilage plate [39].

Score	Morphological Features
0	Absence of rounded chondrocytes in regenerating fibrous cartilage
1	Presence of rounded chondrocytes in regenerating fibrous cartilage
2	Foci of regenerating elastic cartilage with large lacunae surrounded by elastic fibers
3	Continuous area of regenerating elastic cartilage with large lacunae surrounded by elastic fibers

**Table 3 ijms-26-01956-t003:** Scoring system for evaluating elastic fiber content in regenerating cartilage [39].

Score	Morphological Features
0	Absence of elastic fibers
1	Isolated foci (islands) of elastic fibers
2	Elastic fibers close to the edges of the intact cartilage plate
3	Regenerating cartilage plate with elastic fibers

## Data Availability

The data that support the findings of this study are available from the corresponding author upon reasonable request.

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
