# Peer review of "Lactoferrin Stimulates Chondrogenesis and Promotes Healing of the Auricular Elastic Cartilage"

_ijms, 2025, doi:10.3390/ijms26051956_

Round 1

Reviewer 1 Report

Comments and Suggestions for Authors

Thank you for your submission.

This study explores the potential of lactoferrin (LF) to enhance auricular chondrocyte viability, proliferation, and chondrogenesis in both in vitro and in vivo models, showing that LF promotes cartilage regeneration and may benefit tissue engineering and clinical cartilage repair applications. However, there are several issues that need to be addressed by the authors before acceptance.

1.     Could the authors provide an explanation as to why the effects of different concentrations of LF on chondrocyte cell viability do not follow a normal distribution trend?

2.     It seems that the 500 μg/ml concentration has a more significant effect on the cells. Would it be possible for the authors to explain why this concentration was not selected for subsequent experiments?

3.     I noticed that only SOX9 was quantified. It would be helpful if the authors could also consider quantifying COL2, as this would provide a more comprehensive understanding of the effects.

4.     Could the authors please clarify the scale bar in Figure 2E-P and include it in the figure for better clarity?

5.     The connection between the authors' research and tissue engineering is not clearly articulated in the introduction. I suggest that the authors refer to the following articles to help strengthen this section: DOI: 10.1016/j.jot.2024.02.005, DOI: 10.12336/biomatertransl.2024.03.003, DOI: 10.12336/biomatertransl.2023.02.005.

6.     The LF dosage used in the animal experiments seems unrelated to the concentrations used in the in vitro studies. It would be beneficial if the authors could provide clarification on how these dosages were selected.

7.     The study could benefit from further exploration of LF’s impact on the generation of elastic cartilage. A more in-depth investigation into this aspect would add valuable insights to the manuscript.

Author Response

Dear Reviewer,

We sincerely appreciate the time and effort you have dedicated to evaluating our manuscript. Your comments and constructive feedback have been invaluable in improving the clarity, depth, and overall quality of our work. Below, we provide detailed responses point-by-point

  1. Could the authors provide an explanation as to why the effects of different concentrations of LF on chondrocyte cell viability do not follow a normal distribution trend?

Thank you for your insightful comment. We hypothesize that the effects of lactoferrin are complex due to its interaction with multiple receptor types, including TGF-β receptors. According to the literature, the TGF-β signaling pathway is highly unpredictable, as its effects are not solely determined by ligand concentration but also by the balance of intracellular mediators. These mediators can be influenced by various factors, including negative feedback loops and receptor endocytosis. The molecular mechanisms underlying LF’s effects require further investigation, involving proteomic and transcriptomic analyses combined with bioinformatics approaches. We have incorporated this discussion into the revised manuscript (page 7, lines 219–237).

  1. It seems that the 500 μg/ml concentration has a more significant effect on the cells. Would it be possible for the authors to explain why this concentration was not selected for subsequent experiments?

Thank you for your question. Despite its significance, the effect size (Cohen's d) remains moderate in both cases (0.62 vs. 0.79), with the observed difference likely driven by lower variability rather than a higher median. Moreover, previous studies have shown that high doses of LF may exert pro-apoptotic effects (https://pubmed.ncbi.nlm.nih.gov/26996464/), which is why we aimed to avoid dose escalation and selected the minimally effective concentration. It is also worth noting that several studies on articular cartilage (references 16–26) have utilized LF concentrations ranging from 10 to 200 µg/mL, and we took these findings into account in our research. The corresponding revisions have been made on page 3, lines 99–100, and to the Discussion (page 7, lines 205–218, 232–236).

  1. I noticed that only SOX9 was quantified. It would be helpful if the authors could also consider quantifying COL2, as this would provide a more comprehensive understanding of the effects.

We appreciate this valuable suggestion. We quantified type II collagen levels using fluorescence intensity analysis of pellet cultures. This method, previously described in https://pubmed.ncbi.nlm.nih.gov/29955624/, is not as precise as Western blotting for absolute quantification. However, it allows for reliable group comparisons when imaging and staining conditions are standardized. This analysis accounts for both the total stained area and fluorescence intensity, which can indirectly reflect collagen levels. We have included these data in Figure 1G, page 5, lines 128–132) and updated the discussion accordingly to address this aspect as well as the rationale for our concentration selection.

  1. Could the authors please clarify the scale bar in Figure 2E-P and include it in the figure for better clarity?

We apologize for the unclear scale bars. They have been revised and updated in Figure 2.

  1. The connection between the authors' research and tissue engineering is not clearly articulated in the introduction. I suggest that the authors refer to the following articles to help strengthen this section: DOI: 10.1016/j.jot.2024.02.005, DOI: 10.12336/biomatertransl.2024.03.003, DOI: 10.12336/biomatertransl.2023.02.005.

Thank you for highlighting this important aspect. We have expanded the discussion on the potential application of tissue engineering approaches for auricular cartilage regeneration and incorporated the suggested references. These revisions have been made on page 2, lines 48–50.

  1. The LF dosage used in the animal experiments seems unrelated to the concentrations used in the in vitro studies. It would be beneficial if the authors could provide clarification on how these dosages were selected.

We appreciate this observation. Directly translating in vitro concentrations to in vivo studies is challenging due to differences in bioavailability. Therefore, the in vivo dosage was chosen based on previously reported in vivo studies utilizing LF. We have added a corresponding note and references in the Materials and Methods section (page 10, lines 352–353).

  1. The study could benefit from further exploration of LF’s impact on the generation of elastic cartilage. A more in-depth investigation into this aspect would add valuable insights to the manuscript.

Thank you for your recommendation. Indeed, our study primarily focuses on morphological aspects, which is a limitation. However, the mechanisms underlying LF’s role in epimorphic regeneration are complex and require separate investigation. According to the literature, TGF-β signaling and Bcl-2 family proteins, which regulate apoptosis, play crucial roles in auricular cartilage regeneration. LF has been shown to influence both pathways, suggesting a potential mechanism for its pro-regenerative effects. This topic could serve as a foundation for future studies exploring the deeper molecular mechanisms of LF-mediated cartilage regeneration. We have added this discussion to the manuscript (page 8, lines 263–268).

We sincerely appreciate your constructive feedback, which has helped us refine and improve our manuscript.

With best regards,

Anastasiia Kurenkova

Reviewer 2 Report

Comments and Suggestions for Authors

The objectives of the abstract are overly extensive. Please condense them while ensuring that key study details are highlighted. The abstract should be structured according to the authors' guidelines, including a background, objectives, methods, results, and conclusions. Incorporate specific numerical results with corresponding p-values to enhance the rigor and informativeness of the summary.

In the introduction, please provide a citation for the initial paragraph. Additionally, while the auricle plays an important role in sound collection and localization, injuries or malformations primarily affect its aesthetic appearance. In contrast, significant functional impairments of hearing are typically associated with damage to the external auditory canal, middle ear, or inner ear structures. Ensure that this distinction is clearly articulated and supported by appropriate references.

The final paragraph of the discussion is currently lacking a statement of the study's aim. Not just hypothesis but the aim. 

Discussion: Alibegović (2024) demonstrated that semi-automatic image analysis enhances objectivity and reduces variability in the evaluation of cartilage staining. Include this citation within the discussion and acknowledge that the histological evaluations here were not performed (semi)quantitatively. DOI: https://doi.org/10.5566/ias.3171  

Address the potential biases or limitations in the study, including the use of a rabbit model and its translational relevance to human cartilage. Explicitly discuss how species differences might affect the applicability of the findings.

Finally, address minor grammatical inconsistencies and improve readability by simplifying unnecessarily complex sentences. For example, abbreviations such as "LF" and "lactoferrin" should be consistently formatted throughout the text to avoid confusion. Ensure that all terminology and formatting align with the journal's style requirements.

Author Response

Dear Reviewer,

We would like to thank you for thoughtful and detailed comments, which have helped us refine our manuscript. We have carefully addressed each concern and incorporated the necessary changes to enhance the clarity and scientific rigor of our study. A point-by-point response to all comments is provided below.

  1. The objectives of the abstract are overly extensive. Please condense them while ensuring that key study details are highlighted. The abstract should be structured according to the authors' guidelines, including a background, objectives, methods, results, and conclusions. Incorporate specific numerical results with corresponding p-values to enhance the rigor and informativeness of the summary.

Thank you for pointing out this issue and for your valuable suggestions to improve the abstract. We have revised the text by slightly condensing the objectives, expanding the methods section, and incorporating numerical results to enhance clarity and rigor. The changes have been made on page 1, lines 14–27.

  1. In the introduction, please provide a citation for the initial paragraph. Additionally, while the auricle plays an important role in sound collection and localization, injuries or malformations primarily affect its aesthetic appearance. In contrast, significant functional impairments of hearing are typically associated with damage to the external auditory canal, middle ear, or inner ear structures. Ensure that this distinction is clearly articulated and supported by appropriate references.

We greatly appreciate this relevant comment. We have reworded the paragraph to clarify that auricular damage primarily affects sound localization rather than overall hearing function. Additionally, we have included an appropriate citation to support this statement. The changes have been made on page 1, lines 33–35.

  1. The final paragraph of the discussion is currently lacking a statement of the study's aim. Not just hypothesis but the aim.

Thank you for this important observation. In accordance with the journal's guidelines, we have moved the final paragraph to the conclusion section, which follows the methods section. There, we explicitly restate the study's aim, summarize the key findings, and discuss potential applications of our results. The changes have been made on pages 11–12, lines 423–442.

  1. Discussion: Alibegović (2024) demonstrated that semi-automatic image analysis enhances objectivity and reduces variability in the evaluation of cartilage staining. Include this citation within the discussion and acknowledge that the histological evaluations here were not performed (semi)quantitatively. DOI: https://doi.org/10.5566/ias.3171

We sincerely appreciate this valuable suggestion. We acknowledge that the absence of semi-automatic evaluation is a limitation of our study, and we have now explicitly stated this on page 8, lines 272–275. However, it is important to note that Alibegović et al. reported a high correlation between the visual scoring method and semi-automatic image analysis, suggesting that the choice of approach likely did not significantly influence the results. To minimize artifacts related to subjective pathologist assessment, our protocol ensures that animal group assignments are not labeled on the slides, with only the specimen number indicated.

  1. Address the potential biases or limitations in the study, including the use of a rabbit model and its translational relevance to human cartilage. Explicitly discuss how species differences might affect the applicability of the findings.

Thank you for this insightful comment. Indeed, there are some anatomical and biochemical differences between human and rabbit auricular cartilage, although these differences are not drastic. We recognize the inherent limitations of animal models in translating findings to human applications; however, such studies remain essential for developing new treatments. We have expanded the discussion on this limitation in detail. The changes have been made on page 8, lines 276–292.

  1. Finally, address minor grammatical inconsistencies and improve readability by simplifying unnecessarily complex sentences. For example, abbreviations such as "LF" and "lactoferrin" should be consistently formatted throughout the text to avoid confusion. Ensure that all terminology and formatting align with the journal's style requirements

We have carefully rechecked the grammar with the help of editors, split long sentences to improve readability, and ensured consistency in abbreviations and terminology throughout the text, aligning with the journal’s style guidelines.

Thank you for your valuable feedback and the opportunity to improve our manuscript!

With best regards,

Anastasiia Kurenkova

Reviewer 3 Report

Comments and Suggestions for Authors

This work reports the influence of lactoferrin on cell proliferation and wound healing.

At its current stage, the manuscript cannot be accepted for publication due to the following concerns:

- The authors should clarify the methodology and rationale behind the analysis of short- and long-term viability. The study does not clearly define the time points chosen for assessing proliferation, nor the time frame for the SOX9 testing. Many existing studies evaluate proliferation over extended periods (up to 5 days in Petri dish), so further explanation is necessary.

- The comparison of concentrations (10, 100, and 1000 µg/mL) showed in Figure 1G-F is unclear. For instance, Figure 1C highlights that the concentration of 500 µg/mL is statistically different, but this is not adequately discussed.

- Figures 1P-W do not display a clear distinction between the groups. Based on the results, the optimal choice of 10 µg/mL is not well-supported, as it only demonstrates slightly higher proliferation compared to other conditions.

- To better estimate the concentration of lactoferrin with the highest chondrogenic potential, a PCR analysis after 21 days is recommended.

- Some figures are not cited in the manuscript (e.g., Figure 1F). Please ensure all figures are referenced appropriately.

- The conclusions section is too brief and does not sufficiently address the broader implications of the findings. 

Comments on the Quality of English Language

Please revise the text in terms of orthography

Author Response

Dear Reviewer,

We are grateful for the valuable feedback, which has allowed us to strengthen our manuscript. Your suggestions have helped us clarify key aspects of our methodology, analysis, and interpretation of results. We have carefully revised the manuscript and provide detailed responses to each point below.

  1. The authors should clarify the methodology and rationale behind the analysis of short- and long-term viability. The study does not clearly define the time points chosen for assessing proliferation, nor the time frame for the SOX9 testing. Many existing studies evaluate proliferation over extended periods (up to 5 days in Petri dish), so further explanation is necessary.

Thank you for pointing this out. The long-term cytotoxicity test was conducted the day after Alamar Blue was added to the cells. We did not use longer incubation periods because resazurin has a slight intrinsic cytotoxicity, which could distort the results (https://pmc.ncbi.nlm.nih.gov/articles/PMC4294845/). However, we assessed cell proliferation over three passages (Figure 1F) when cells were transferred to a new vessel, which took approximately three weeks. Regarding the time points for Sox9 quantification in pellets, they were formed over a three-week period, as described in the methodology section. However, we recognize that this may not have been explicitly clear, so we have added the relevant clarification to the immunohistochemical (IHC) analysis section in the Methods (page 10, line 365-366) and the Results (page 5, line 120).

  1. The comparison of concentrations (10, 100, and 1000 µg/mL) showed in Figure 1G-F is unclear. For instance, Figure 1C highlights that the concentration of 500 µg/mL is statistically different, but this is not adequately discussed.

We appreciate the opportunity to clarify our choice of concentrations. Indeed, LF at 10 and 500 µg/mL showed a significant effect. However, we focused on lower concentrations because previous studies suggest that high doses of LF may have a pro-apoptotic effect (https://pubmed.ncbi.nlm.nih.gov/26996464/). Additionally, we referenced prior research on articular cartilage (references 16–26), where authors used concentrations ranging from 10 to 200 µg/mL. The 1000 µg/mL concentration was included as the maximum possible dose, but based on literature data, we did not expect a significant effect at this level. We aimed to confirm that increasing the dose does not necessarily enhance the effect, which was also demonstrated in our study. We have added these explanations to page 3, lines 99–100, and to the Discussion (page 7, lines 205–218, 232–236).

  1. Figures 1P-W do not display a clear distinction between the groups. Based on the results, the optimal choice of 10 µg/mL is not well-supported, as it only demonstrates slightly higher proliferation compared to other conditions.

Thank you for highlighting this issue. We acknowledge that visually assessing the percentage of positive cells or fluorescence intensity can be challenging, which is why we conducted quantitative analysis. In the case of Sox9, the number of positive cells increased by 30% at 10 µg/mL, and for type II collagen, the difference was twofold at 100 µg/mL. This is a substantial effect (Cohen’s d > 2). However, we agree that stating 10 µg/mL as the optimal concentration was not strongly justified. To address this, we have softened our wording throughout the manuscript and now refer to an optimal range of concentrations, with 10 µg/mL being the minimum effective dose required to achieve the observed effects.

  1. To better estimate the concentration of lactoferrin with the highest chondrogenic potential, a PCR analysis after 21 days is recommended.

Due to the rigid structure of the pellets formed using our described method, RNA extraction and PCR analysis from them is technically challenging. Additionally, gene expression does not always correlate directly with protein synthesis, so assessing the protein composition appears more relevant. We evaluated this using immunohistochemistry. However, we acknowledge that Sox9 quantification alone may not be sufficient. To strengthen our analysis, we performed an additional assessment, indirectly quantifying type II collagen using the method previously described in https://pubmed.ncbi.nlm.nih.gov/29955624/. This approach is applicable when staining conditions and imaging settings are standardized, allowing us to assess both the stained area and fluorescence intensity, which reflects the amount of protein present. While this method is not as precise as Western blotting, it is sufficient for comparing groups. These changes are reflected in Figure 1G, page 5, lines 128–132 and the Discussion section, where we revised our argumentation regarding concentration selection.

  1. Some figures are not cited in the manuscript (e.g., Figure 1F). Please ensure all figures are referenced appropriately.

We apologize for this oversight. The reference to Figure 1F has been added on page 2, line 97, and all other figure citations have been carefully reviewed and corrected as needed.

  1. The conclusions section is too brief and does not sufficiently address the broader implications of the findings. 

Thank you for this insightful suggestion. We have expanded the Conclusions section to better address the broader implications of our findings and the future directions of our research. These revisions can be found on pages 11–12, lines 423–442.

We greatly appreciate the reviewers’ constructive feedback, which has helped us significantly improve our manuscript. Thank you for your time and effort in reviewing our work.

With best regards,

Anastasiia Kurenkova

Round 2

Reviewer 1 Report

Comments and Suggestions for Authors This manuscript has improved considerably after revision, which has addressed reviewers' previous concern. I support the acceptance of this work.

Author Response

We would like to once again express our gratitude to the reviewer for the valuable comments, which have significantly improved the manuscript, as well as for the support in its acceptance.

Reviewer 3 Report

Comments and Suggestions for Authors

The authors responded to the reviewer's comments, although some points and answers remain unclear.

1. If the authors believe that resazurin may affect cell viability, the reviewer strongly advises adopting an alternative reagent to assess cell viability, as several commercially available options exist. The current response is considered inadequate.

2. Regarding the effects of LF doses, the authors indicated that high doses may have a pro-apoptotic effect. Specifically, the dose of 500 µg/mL showed a statistically significant difference compared to the control, while the 1000 µg/mL dose yielded results similar to the control. The authors should further substantiate these findings with appropriate statements.

4. The technical justification for not applying PCR is insufficient. Furthermore, a major concern is that the results presented in Graph 1G do not align with the data shown in Images 1Q–1T. Further clarification is required.

Author Response

We thank the reviewer for the thorough evaluation of our work. Below, we address the comments point by point.

  1. If the authors believe that resazurin may affect cell viability, the reviewer strongly advises adopting an alternative reagent to assess cell viability, as several commercially available options exist. The current response is considered inadequate.

According to the manufacturer’s instructions, cell incubation with Alamar Blue should occur for 1–4 hours. This duration is recommended for assessing the toxicity of drugs and materials. The results of viability assessment using resazurin correlate well with viability assays such as LDH, MTT, and CFDA-AM, which employ different mechanisms (https://pmc.ncbi.nlm.nih.gov/articles/PMC2438350/, https://link.springer.com/article/10.1007/s10616-024-00670-x, https://pubmed.ncbi.nlm.nih.gov/15251189/, etc.). Thus, a single assay is sufficient for preliminary screening. For translational research, a broader range of tests will undoubtedly be required, but such studies should be conducted on human cells rather than rabbits. This study focuses exclusively on rabbits.

Additionally, we used a second time point with 17 hours of incubation with resazurin, although this is not a standard approach. This time point was chosen because it allows us to assess the long-term effects of lactoferrin while avoiding the cytotoxic effects of resazurin, which begin to manifest after two days of incubation (https://pmc.ncbi.nlm.nih.gov/articles/PMC4294845/). For an even longer assessment of LF's effects on chondrocytes, we evaluated cell proliferation over three passages, approximately at weeks 1, 2, and 3 of LF cultivation. A significant effect was observed only at week 3.

We have added clarifications regarding LF's duration of action in the Results and Methods sections, as well as updated the study limitations in Discussion regarding future translational research. The added text is given below

Results (page 2, lines 89-91)

«… To evaluate LF long-term impact, we assessed its effects on proliferation. Cells were expanded over three passages, and the fold increase in cell number was recorded as the proliferation rate. …»

Discussion (page 6-7, lines 199-205)

«...It is important to note that many metabolic assays, including Alamar Blue, are not suitable for evaluating the long-term effects of substances due to their intrinsic impact on metabolism [27,28]. Therefore, for periods exceeding 24 hours, we assessed the effect of LF by evaluating cell proliferation. Our research demonstrated that LF does not affect the division of auricular chondrocytes in early passages; however, its effects become more pronounced by the third passage, i.e., by the third week of cultivation…»

Discussion (page 8, lines 288-293)

«...Thirdly, this study was conducted on rabbits, which may influence the results. For future translational research, it is crucial not only to replicate our findings but also to consider additional factors. For instance, we used only the resazurin-based assay for the primary evaluation of cytotoxicity and viability, as its results correlate well with those of other metabolic tests [47]. However, for human applications, a broader panel of assays is required…»

Methods (page 9-10, lines 363-366)

«For short-term viability assessment, cells were cultured for 4 hours, as recommended by the manufacturer. We also attempted to assess longer-term effects and for this, cells were cultured for 17 hours (overnight)»

  1. Regarding the effects of LF doses, the authors indicated that high doses may have a pro-apoptotic effect. Specifically, the dose of 500 µg/mL showed a statistically significant difference compared to the control, while the 1000 µg/mL dose yielded results similar to the control. The authors should further substantiate these findings with appropriate statements.

In our experiments, the 1000 µg/mL concentration did not affect the observed parameters, including pro-apoptotic effects. However, the dose-response relationship of LF is clearly non-linear. Low and very high concentrations do not affect measured parameters, whereas optimal concentrations (10–500 µg/mL) exert stimulatory and pro-chondrogenic effects. Our findings align with existing literature, which suggests that increasing LF concentration may result in a lack of effect or even adverse changes.

To clarify this point, we have revised the Discussion section (page 7, lines 217-221).

«…Notably, increasing the LF concentration to 1 mg/ml did not produce significant effects in either 2D or 3D cultures. The literature indicates that low doses of LF can stimulate metabolism in intestinal epithelial cells, whereas higher doses exhibit pro-apoptotic effects [32]. In our study, 1 mg/ml of LF did not exert such effects; however, the protein activity decreased with increasing concentration…»

  1. The technical justification for not applying PCR is insufficient. Furthermore, a major concern is that the results presented in Graph 1G do not align with the data shown in Images 1Q–1T. Further clarification is required.

The absence of molecular techniques such as PCR is indeed a limitation of our study. Performing this methodology requires a substantial number of pellets due to low RNA yield, which in turn necessitates a larger number of animals to obtain primary cultures. To demonstrate LF’s pro-chondrogenic effect, we conducted immunohistochemical analysis, as tissue composition ultimately determines cartilage type. Many studies on chondrocytes, including auricular chondrocytes, show that changes in cartilage marker expression correlate with protein level alterations (e.g., https://pubmed.ncbi.nlm.nih.gov/29485180/, https://pmc.ncbi.nlm.nih.gov/articles/PMC9322318/, https://pmc.ncbi.nlm.nih.gov/articles/PMC5688648/). Therefore, we do not believe PCR results would contradict our findings. However, we recognize that such data may be relevant for future translational studies and have addressed this in the Discussion section.

Page 7, lines 213-216

«…It is worth noting that we did not perform PCR analysis due to limitations in the amount of biological material from animals, however, earlier studies using chondrocytes show similar patterns between the expression of extracellular matrix proteins and their content in tissue [29–31]…»

page 8, lines 269-275

«…Our study has several limitations that should be addressed in future research. First, although we demonstrated the overall pro-chondrogenic effect of LF in both in vitro and in vivo experiments, we did not explore its molecular aspects in depth. As previously described, the mechanism of LF action may involve modulation of TGF-β signaling pathway activity. Gene expression analysis or quantification of specific molecules in this cascade could provide further insights into both the stimulation of proliferation and the effects of LF on chondrogenesis markers…»

page 8, lines 293-295

«…Additionally, while we identified an optimal range of LF concentrations for future studies, selecting a dose for human application should consider a more extensive set of markers related to elastic tissue and regeneration, including their expression levels…»

Page 9, lines 313-218

«…Thus, within the framework of this study, we have confirmed the hypothesis that LF can influence the regeneration of elastic cartilage. Future translational studies employing a broader range of methods, such as comprehensive cytotoxicity assessment, in situ hybridization, and spatial transcriptomics, will provide insights into the fundamental mechanisms of organotypic regeneration and the activation of these mechanisms in differentiated tissues of adult organisms…»

Regarding the figure, we analyzed data from four pellets and acknowledge that we selected images based on size similarity rather than representativeness. We have corrected this oversight by replacing the images with those that visually align better with the results. These changes have been made to Figure 1.

Once again, we sincerely thank the reviewer for the opportunity to improve our work, resolve potential misunderstandings, and clarify key aspects.